# Biocontrol of *Botrytis cinerea* as Influenced by Grapevine Growth Stages and Environmental Conditions

**DOI:** 10.3390/plants12193430

**Published:** 2023-09-28

**Authors:** Valeria Altieri, Vittorio Rossi, Giorgia Fedele

**Affiliations:** Department of Sustainable Crop Production (DiProVeS), Università Cattolica del Sacro Cuore di Piacenza, 29122 Piacenza, Italy; valeria.altieri@unicatt.it (V.A.); vittorio.rossi@unicatt.it (V.R.)

**Keywords:** biological control, biological control agents, decision making, integrated pest management

## Abstract

The growth of four commercial biocontrol agents (BCAs: *Bacillus amyloliquefaciens* (BAD), *Aureobasidium pullulans* (APD), *Metschnikowia fructicola* (MFN), and *Trichoderma atroviride* (TAS)) was evaluated using turbidimetric assays on artificial substrates mimicking the chemical berry composition at four stages: pea-sized berries, veraison, softening, and ripe berries. The response of BCA growth differed among BCAs. Subsequently, the BCAs’ population size was assessed after 1 to 13 days of incubation on the substrate mimicking ripe berries at 15 to 35 °C. The population size of BAD increased with temperatures, while that of MFN decreased; the population sizes of APD and TAS showed bell-shaped patterns with lower growth at 15 or 35 °C. Finally, the BCAs were applied to ripe berries and then incubated at 15 to 30 °C. After 1 to 13 days, the berries were inoculated with *B. cinerea* and incubated for 7 days, after which the BCA control efficacy was assessed. The highest control was observed at 25 °C for BAD and APD, at 15 to 20 °C for MFN, and at 25 to 30 °C for TAS. The results confirm that the plant substrates and temperature affect the population size of the BCA following application; temperature also affects the preventative efficacy of BCA against *B. cinerea*.

## 1. Introduction

*Botrytis cinerea* Pers.:Fr (anamorph of *Botryotinia fuckeliana* (de Bary) Whetzel) is the causal agent of gray mold, an important disease that affects grapevine (*Vitis vinifera* L.) and causes significant yield and quality losses worldwide [1]. *B. cinerea* develops as a saprophyte, necrotroph, or parasite on multiple grape organs including leaves, green shoots, rachides, flowers, bunch trash (such as calyptras, dead stamens, aborted flowers and berries, and tendrils), and ripening berries. *B. cinerea* also has multiple infection pathways, and infections mainly occur from flowering to fruit set and after veraison [1,2].

Due to the economic importance of the disease, gray mold control is traditionally based on a routine application of synthetic fungicides at four grape growth stages (GSs): (1) flowering (A; GS65; [3]), (2) pre-bunch closure (B; GS77), (3) veraison (C; GS83), and (4) before harvest (D; before GS89), as described by González-Domínguez et al. [4]. This application scheduling provides effective disease control [5,6,7] but often leads to unnecessary spraying [8], causes an increase in the risk that the pathogen will develop fungicide-resistant populations [9,10], may leave fungicide residues on grapes and wines [11], and contributes to the intensive use of chemicals in viticulture [12] with consequent negative impacts on human health and the environment [13,14]. Plant protection products of natural origin are then viewed as alternatives that can substitute or complement the use of synthetic fungicides for gray mold management [15,16,17].

Natural products include microbial biocontrol agents (BCAs), such as yeasts, fungi, and bacteria [18,19]. Biocontrol can be defined as the use of microbial strains that reduce the incidence or severity of crop diseases [20] via the suppression of the pathogen through different modes of action (MoA), which include competition for nutrients and space, antibiosis, parasitism, and induced host plant resistance [15,18,19]. BCAs have a small impact on human health and the environment, are useful in anti-resistant strategies because they show different MoA with respect to synthetic fungicides [21], and have a short pre-harvest interval (Phi). In recent years, the number of BCAs available for gray mold control has increased [22,23]. Compared to synthetic fungicides, however, BCAs often show lower and more variable efficacy than fungicides when applied under field conditions [17]. The successful application of BCAs should differ from the routine application of synthetic fungicides and be based on several steps: (1) an evaluation of the risk of infection; (2) a consideration of the target infection pathway of *B. cinerea* and the consequent plant substrate on which the BCA should grow; and (3) a consideration of weather conditions at the time of application and after, which affect BCA survival, growth, and efficacy.

For the evaluation of the risk of infection by *B. cinerea* (step 1), mathematical models have been proposed [24,25,26]. In recent years, a mechanistic, weather-driven model, which accounts for different infection pathways of *B. cinerea*, the host growth stage, and the weather conditions, was developed and validated and provides an accurate prediction of gray mold severity on a daily basis [4,27]. For the consideration of the target infection pathway of *B. cinerea* (step 2), Elmer and Michailides [1] defined the different infection pathways of *B. cinerea* on grapes from which the required activity of BCAs can be drawn. From pre-flowering to fruit set, the BCA should prevent both conidial germination on inflorescences and the latent infection of berries (infection pathways I, IIa, and IIb of Elmer and Michailides [1]). From fruit set to pre-bunch closure, the BCA should colonize bunch trash (infection pathway III) and prevent sporulation by *B. cinerea* (infection pathway IV). From veraison until harvest, the BCA should colonize the berry surface and prevent infection by conidia (infection pathway Va) and aerial mycelium (berry-to-berry infection, pathway Vb; [4]).

Plant substrates on which the BCAs should grow then include flowers and flower parts, bunch trash, and berries at different ripening stages [28,29]. The biochemical characteristics (including pH, nutrient types and availability, and water potential) of these plant substrates differ from one to another, as well as varying during the season, and that variation affects both the susceptibility of grapevine to *B. cinerea* infection, such as an increase in berry susceptibility as ripening advances [29,30], and the capability of BCAs to profusely grow and colonize the plant substrate. Our knowledge about the capability of different BCAs to colonize different plant substrates, however, is still limited [31].

For the consideration of the effects of weather conditions on BCA survival, growth, and efficacy (step 3), it was previously demonstrated that temperature and humidity not only affect the mycelial growth, conidial germination, and sporulation of *B. cinerea* [30,32] but also affect BCA survival, growth, and efficacy [33,34,35]. It was also demonstrated that rain washes off the microbial populations on the berry plant surface [36] and produces a reduction in BCA efficacy [37]. As for the capability of BCAs to colonize different plant substrates, our knowledge concerning weather effects on BCA is still limited, and further research is needed.

We conducted three experiments to contribute to our knowledge about the effects of four commercial biocontrol products (each with different BCAs) on artificial substrates that mimic the chemical composition of grape juice at four berry ripening stages, on BCA growth at five temperature regimes on an artificial substrate mimicking ripe berries, and on the BCA efficacy in controlling gray mold at four temperature regimes on ripe berries when applied 1 to 13 days before *B. cinerea* infection.

## 2. Results

### 2.1. Effect of Berry Growth Stage on the BCA Growth

When the four BCAs were grown for 48 h at 20 °C on artificial substrates mimicking the chemical composition of the berries at different growth stages, the area under the growth curve (AUGC) was significantly affected by the BCA and berry growth stage (*p* < 0.001), which accounted for 68.0% and 5.4% of the total variance in AUGC, respectively. The interaction was also significant (*p* < 0.001) and accounted for 26.5% of the total variance, indicating that the BCA growth was different for different berry growth stages (Figure 1). *Bacillus amyloliquefaciens* subsp. plantarum (BAD) grew more at GS83 and GS85 than at earlier or later stages (Figure 1A), and *Aureobasidium pullulans* (APD) grew more at GS83 than at the other stages (Figure 1B). The growth of *Metschnikowia fructicola* (MFN) increased as berry ripening progressed (Figure 1C), while *Trichoderma atroviride* (TAS) did the opposite (Figure 1D). The coefficient of variations (CVs) of AUGC indicated that the growth of TAS (CV = 146.5%) was more sensitive to variation in the berry growth stage than the other BCAs and, particularly, than BAD (CV = 71.7%), as shown in Figure 1.

When the four BCAs were grown on artificial substrates that mimicked the chemical composition of ripe berries (GS89), the number of CFUs was significantly affected by the BCA, temperature, and length of the colonization period (*p* < 0.001) and accounted for 27.9%, 10.8%, and 5.4% of the total variance in CFUs, respectively. The interaction ‘BCA × temperature × colonization period’ was also significant (*p* < 0.001) and accounted for 9.4% of the total variance, indicating that the number of CFUs was influenced by both the temperature and length of the colonization period. BAD produced the highest CV of relative population size (103.4%) followed by MFN (77.5%), APD (59.5%), and TAS (46.8%). The relative population size of BCAs as affected by length of the colonization period and temperature is shown in Figure 2. Some BCAs showed faster growth than others (Figure 2A). For instance, the maximum population size of MFN and TAS were observed at 3 and 9–13 days, respectively. The response of the BCA population size to temperature was also different (Figure 2B). The population size of BAD increased with temperature from 15 to 35 °C, while that of MFN decreased, and the population size of APD and TAS showed bell-shaped patterns with lower growth at 15 or 35 °C (Figure 2B).

Figure 3 shows the effect of interaction ‘BCA × temperature × colonization period’ on the relative population size of BAD (Figure 3A) and MFN (Figure 3B). At 25, 30, and 35 °C, BAD population increased until the ninth day of the colonization period, with the maximum at 35 °C after 9 days. No increases were observed at 15 and 20 °C (Figure 3A). At 15 and 20 °C, the MFN population decreased after 6 days of the colonization period while at 25, 30, and 35 °C, the decrease was observed after 3 days; the maximum occurred at 15 °C after 6 days (Figure 3B).

### 2.2. Effect of Temperature on the BCA Efficacy on Ripe Berries

When the efficacy of the four BCAs was evaluated on field-collected, ripe berries, a significant effect of the BCA (*p* < 0.001) and the length of the colonization period (*p* = 0.006) was found to have accounted for 30.2% and 11.3%, respectively, of the total variance in relative control. The interaction ‘BCA × temperature’ was also significant (*p* = 0.008) and accounted for 17.3% of total the variance, indicating that the BCA efficacy was influenced by temperature. The CV of relative control indicated that the efficacy of MFN (CV = 53.3%) was more sensitive to the temperature and length of the colonization period than the other BCAs (BAD = 25.9%, APD = 37.9%, and TAS = 27.7%). Relative gray mold control by BCAs as affected by the length of BCA colonization period and temperature is shown in Figure 4. Some BCAs were effective more rapidly than others (Figure 4A). For instance, the relative control of MFN was 0.80 ± 0.06 after only 1 day post-BCA inoculation, while BAD and TAS required 9 days until they showed a similar effect of relative efficacy. Although the BCA responses of gray mold control to temperature had bell-shaped patterns, the BCAs differed in the size of response to specific temperature (Figure 4B). The maximum relative control occurred at 25 °C for BAD and APD, at 15 to 20 °C for MFN, and at 25 to 30 °C for TAS (Figure 4B).

## 3. Discussion

In the present work, we studied four commercial BCAs for control of *B. cinerea* in vineyards that differ in terms of their MoAs. *Bacillus amyloliquefaciens* subsp. plantarum strain D747 (BAD) is a spore-forming bacterium whose major MoA is antibiosis through the production of lipopeptide antibiotics, antifungal proteins, lytic enzymes, and volatile compounds [19,38,39,40]. *Aureobasidium pullulans* strains DMS 14,941 and 14,940 (APD) are yeast-like fungi that grow epiphytically on grape berries and colonize cracks on the berry surface. They produce hydrolytic enzymes and volatile organic compounds and form biofilms [41,42,43,44], after which they exert an antagonistic activity and compete with pathogens for nutrients and space [40,43,45]. As for APD, the primary MoA of the yeast *Metschnikowia fructicola* NRRL Y-27328 (MFN) is competition for nutrients and space via biofilm formation [46,47]. *M. fructicola* can also antagonize fungal and bacterial growth via iron depletion [48]. Different authors reported additional MoAs for yeasts, such as enzyme secretion, parasitism, and the production of volatile organic compounds (VOCs) and metabolites [46,49], but no information exists about the expression of these MoAs by *M. fructicola* NRRL Y-27328. *Trichoderma atroviride* strain SC1 (TAS), along with other *Trichoderma* species, shows multiple MoAs, including mycoparasitism, antibiosis, competition for space and nutrients, and the induction of plant resistance [50,51].

Although MoA is likely to be of primary importance for *B. cinerea* biocontrol, the selection of the BCA based on the MoA may not be sufficient for ensuring high biocontrol efficacy [19]. In addition to MoA, the *B. cinerea* infection pathway to be targeted, the grape growth stage, and the weather conditions are key factors for ensuring BCA survival and colonization on the plant substrate and effective gray mold control [52].

In this work, we focused on the effect of BCAs during berry ripening, which is a key period for the control of gray mold [8]. In this period, berries carrying latent infection can produce typical gray mold rot under favorable weather conditions and affect adjoining berries through contact (berry-to-berry infection pathway). Berries can also be affected by conidia that germinate on the berry surface and produce hyphae that grow epiphytically and penetrate through microcracks or wounds on the berry skin (the conidial infection pathway). From veraison to ripening for harvest, the berries undergo profound changes that involve pulp composition and hardness, the characteristics of the berry skin, and the chemical composition of berry exudate, which represents a source of stimulatory/nutritional compounds for the epiphytic microorganisms, including *B. cinerea* [53,54,55] and BCAs [56]. Ciliberti et al. [30,32] demonstrated that pathogen inoculum production, conidia germination, and colony growth are strongly influenced by the grape berry ripening stage; specifically, a higher sporulation level was observed at the softening berry growth stage (GS85), while the lowest rates of germination were obtained globally at pea-sized (GS75), and the best growth was reported at ripe berries (GS89).

The effect of the berry ripening stage on the growth of BCAs, however, has been less investigated than other stages. Therefore, we evaluated the effect of four berry ripening stages on the growth of the four BCAs after 48 h of incubation in liquid substrates mimicking the berry’s composition at the different stages [30]. We decided to conduct our experiments with artificial substrates instead of using field-collected berries so that our experiments would be reproducible because the chemical composition of the berries may be highly variable in different vineyards and cluster positions on the vine [57,58]. To our knowledge, this study is the first time these substrates instead of generic substrates, such as PDA used for fungi [59,60] or nutrient broth for bacteria [61], were used to study BCA growth. In a previous study, Carbò et al. [62] studied the growth of *Candida sake* on a synthetic grape juice substrate, but this substrate did not correspond to any specific berry growth stage. It can be questioned whether our artificial substrates represent the real conditions for BCA growth on the berry surfaces. It is known that the chemical composition of the berry surface strongly depends on the composition of the pulp, with the compounds from the inner cell layers exudating through the cuticles and epicuticular waxes or flowing out through microcracks and wounds [56,63,64]. It is also known that the exudates’ chemical composition affects the microorganisms present on the grape berry surface [56]. Therefore, the chemical composition of the berry surface mirrors the chemical characteristics of the grape juice and their changes during ripening; for instance, malic acid concentration decreases as ripening advances, while sugar concentration increases rapidly in the later stages of ripening [63]. Since our artificial substrates mimic the chemical composition of the berry juice and its change over time [65], we may then speculate that they mirror the chemical composition of the berry surface.

We evaluated the BCA growth by measuring the OD of the different substrates after 48 h of incubation. Turbidimetric assays have been mainly used for the quantification of microbial population growth in real time when the incubation times are short [62,66,67]. Our results show that some BCAs grow better at later and others at earlier berry ripening stages. This result was expected because each microorganism has specific nutritional requirements [68]. In previous studies, carbon and nitrogen sources, carbon concentrations, and C/N ratios for mycelial growth and sporulation were substantially different among different biocontrol fungi [69,70]. This finding can have relevant practical implications, indicating that the berry ripening stage at the time of BCA application in vineyards may be considered when selecting the BCA to be applied. Further studies are needed to confirm this finding using in-vineyard studies.

In addition to the berry ripening stage, we also considered the effect of temperature on the growth of the four BCAs on ripe berries (a substrate mimicking GS89). In this study, we assessed the BCA growth as CFUs instead of ODs (as in the previous study) because we were interested in the dynamics of living microorganisms over a longer time (until 13 days after application). Indeed, CFUs better express the balance between microorganism growth and death at any time during the study with respect to optical measurements [71]. Living cells of some BCAs were more abundant at 15 °C (MFN) and others at 35 °C (BAD) and in early MFN or late BAD colonization stages. In a previous study, we evaluated the combined effects of temperature and humidity and the colonization period length on BCA growth on grape bunch trash [28]. The responses to temperature of the four BCAs presented in both studies ([28] and this study) were consistent, indicating that the temperature responses of microorganisms did not change depending on the substrate (bunch trash or ripe berries); otherwise, the dynamics of BCA growth over the colonization period were different. For example, APD required 3 and 13 days at 25 °C for the maximal growth on bunch trash and ripe berries, respectively. It seems likely that the nutritional compounds on the surface of ripe berries supported longer growth in microorganisms than the compounds on the dead material of bunch trash.

In parallel to the BCA living populations, we considered the effects of temperature and the length of the colonization period on the efficacy of gray mold control for which ripe berries were treated with single BCAs and inoculated with conidia of *B. cinerea* at different times after BCA application. Artificial inoculation has often been used to study relationships between microbial agents and target pathogens [72,73] because it makes it possible to consider specific factors while maintaining others in a constant state. In these studies, the disinfection of berry surfaces eliminated epiphytic microbial populations that could differ among seasons and vineyards [57,58], and artificial inoculation eliminated the effect of varying inoculum doses and timing on *B. cinerea* infection [74]. Artificial inoculation, however, often leads to efficacy levels that are not comparable with those that can be obtained under field conditions [18]. In our work, however, we were interested in the comparison of the efficacy under different conditions of temperature and colonization time of each BCA rather than comparing BCAs for their efficacy or inferring their efficacy under field conditions. For this reason, we expressed efficacy as relative efficacy rather than using raw efficacy data [29].

Our results indicate that gray mold control by BCAs was significantly affected both by temperature and by the length of the BCA colonization period, with some BCAs being more effective at lower temperatures than others and with shorter colonization periods than others. The efficacy response to both temperature and time, however, was not in line with the results on the dynamics of living BCA populations. For instance, gray mold control by BAD was higher at 25 °C than at 30 °C, while CFUs were higher at 30 °C than at 25 °C. Disease control by TAS was similar at 25 and 30 °C, while CFUs were much lower at 30 °C than at 25 °C. It may be then argued that conditions for BCA growth and efficacy against *B. cinerea* may differ as described in a few previous studies. For instance, the optimal temperature for growth was higher than the one for the production of fungi toxic metabolites by *Trichoderma* spp. [75], and Polizzi et al. [76] reported the effects of temperature on the production of microbial volatile compounds by *T. atroviride*. In general, when a BCA has multiple MoAs, such as for *Trichoderma* species, the expression of each MoA is influenced by several factors (namely soil type, temperature, pH, moisture, and other members of the microflora) as described by Howell [77]. For instance, temperature has a profound effect on the production and activities of enzymes and antibiotics associated with biocontrol by *Trichoderma* spp. [77]. Similarly, *Bacillus* spp. produce antifungal substances [78,79,80,81], and specifically, *B. amyloliquefaciens* (the microorganism of BAD) affects *B. cinerea* through the production of lipopeptide antibiotics, antifungal proteins, lytic enzymes, and volatile compounds [19,38,39,40] whose expression was demonstrated to be influenced by some abiotic factors, including temperature [82].

Overall, our results confirm that the characteristics of the plant substrate on which the BCA should grow (specifically, of the berries at different ripening stages) and temperature both affect the population size of the BCA following application. Temperature also affects the preventative efficacy of BCA against *B. cinerea*. Our results then strengthen the idea that multiple criteria should be considered for the selection of the BCA for a specific application in the vineyard. For instance, when a model-based risk of infection by *B. cinerea* at berry softening in GS85 is present, and an intervention is needed with a BCA that can rapidly colonize the berry surface and prevent infection by conidia, BAD and MFN should be preferred (Figure 2) with the choice of MFN as preferable in cases where temperatures at the time of application and in the following days (based on weather forecasts) are <25 °C (Figure 2B and Figure 3B). If a BCA intervention is needed to protect berries in early ripening stages even though an incoming infection risk is not present, TAS should be preferred because of its capability to colonize berry surfaces at those stages (Figure 2) and its capability to progressively cause an increase in both the population size over time (Figure 2A) and gray mold control (Figure 4A) over a wide temperature range (Figure 4B). These examples, however, need to be confirmed with in-vineyard experiments.

## 4. Materials and Methods

### 4.1. Biocontrol Agents

Four commercial BCAs were used (Table 1), representing different kinds of microorganisms and characterized by different prevalent mode of action (MoA). These products were dispersed in double-distilled sterile water (pH 6.5) at the label dose. The viability of each BCA was confirmed by plating the product suspensions on potato dextrose agar ([PDA] Biolife Italiana S.r.l., Milano, Italy).

### 4.2. Effect of Berry Growth Stage and Temperature on BCA Growth

Four liquid substrates that mimic the chemical composition of grape berries at (1) pea-sized berries (GS75; [3]), (2) veraison (GS83), (3) softening of berries (GS85), and (4) ripe berries (GS89) were prepared as described by Ciliberti et al. [30]. Briefly, substrates were prepared by adding different quantities of sugars (glucose and fructose; Carlo Erba Reagents), organic acids (malic and tartaric acid; Carlo Erba Reagents), and salts (ammonium sulfate, ammonium dihydrogen phosphate, monopotassium phosphate, and magnesium sulfate; Carlo Erba Reagents) to double-distilled water. The pH of the substrates was adjusted (to 2, 2.5, 3, and 3.5 for GS75, GS83, GS85, and GS89, respectively) using potassium hydroxide or phosphoric acid (Carlo Erba Reagents) after autoclaving. The artificial substrates were used as the growing substrates for BCAs. In our experimental setting, these substrates reproduced the chemical composition of the berry surface on which the BCAs grow. These substrates are strongly influenced by grape juice exudates that reach the berry’s surface through cuticles and the epicuticular wax layer or through microcracks [63].

In the first experiment (Exp1), the four artificial substrates were inoculated with a suspension of each BCA in 50 mL Falcon tubes (1:10, *v*/*v*, of BCA suspension/artificial substrate) at the label dose of Table 1, and vortexed for 2 min. To determine the growth of BCAs on the liquid substrates, aliquots of 200 µL of substrate were then loaded into 96-well plates using three-well plates for each combination ‘substrate × BCA’. The BCA population size was determined as the turbidity of the suspension (optical density [OD]), which has been previously used to monitor microbial growth [62]. A BioTek 800 TS absorbance reader (Agilent Technologies, Santa Clara, CA 95051, USA) was operated for 48 h, and the optical density was recorded every 2 h using a 600 nm filter (OD_600_) and the software Gen5 provided by the manufacturer. Plates were shaken at 567 cpm linear frequency for 30 s before every automatic reading, and a total of 25 reads for each well was obtained (from t_0_, the initial reading to t_25_, the final reading after 48 h from the starting time. Readings were obtained at 2 h intervals). Non-inoculated artificial substrates were used to measure substrate turbidity (control). Before further analysis, the raw turbidity data were corrected for the turbidity of the control: t_n_ − t_0_, with t_n_ from t_1_ to t_25_. The experiment was performed twice.

BCA growth was finally assessed by calculating the area under the growth curve (AUGC):$$\mathrm{AUGC}=\sum_{t-1}^{Nt-1}\frac{\left(y_{i}-y_{i+1}\right)}{2}{(t}_{i+1}-t_{i})$$
in which y_i_ − y_i+1_ is the sum of two consecutive values of OD_600_, and t_i+1_ − t_i_ is the 2 h interval between two consecutive measurements. The final values of AUGC were subject to factorial analysis of variance (ANOVA) in which the factors included BCA treatment (four BCAs) and berry composition (four artificial substrates: GS75, 83, 85, and 89). The experimental design was a split plot with BCAs as the main plot and artificial substrates as the split plot. The two repetitions of the experiment were considered random factors. The Tukey’s Honest Significant Difference test was used at α = 0.05 to discriminate between AUGC means. To assess the variability in the AUGC as affected by the berry growth stage, coefficients of variation (CV, in %, or relative standard deviations) were used. A higher CV indicated a greater variability generated by the considered factors.

In a second experiment (Exp2), the artificial substrate mimicking the chemical composition of ripe berries (GS89) was inoculated with each BCA as previously described in 1.5 mL tubes, and tubes were then incubated at 15, 20, 25, 30, or 35 °C in a growth chamber with 100% relative humidity and 12 h photoperiod. After 1, 3, 6, 9, and 13 days, the colony forming units (CFUs) per mL of suspension were determined by serially diluting the inoculated suspensions after which 100 µL aliquots of each serial dilution were plated on PDA plates (9 cm in diameter) and incubated at 25 °C for 48 h to evaluate the BCA population size over time at different temperatures. There were three tubes (replicates) for each temperature and incubation time. The experiment was performed twice.

The CFU data were subjected to factorial analysis of variance (ANOVA), in which the factors were BCA treatment (four BCAs), temperature regime (15, 20, 25, 30, and 35 °C), and the number of days of incubation (1, 3, 6, 9, or 13 days); the experimental design was a split-split, with BCAs as the main plots, temperature regimes as the split plots, and BCA colonization periods as the split-split plots. The two repetitions of the experiment were considered random factors. The CFU data were subjected to ln(x+1) transformation before applying the ANOVA so that variances were homogeneous. To assess the variability in the number of CFUs as affected by temperature and incubation time (which is the colonization period), CVs were calculated as described before.

### 4.3. Effect of Temperature on the BCA Efficacy on Ripe Berries

Grape berries were collected in 2018 and 2019 in an 11-year-old vineyard (in 2018) located at Castell’Arquato (44°51′26.1′′ N 9°51′20.7′′ E, 400 m a.s.l.) in the Emilia-Romagna Region. The vineyard was planted with cv. Merlot (highly susceptible to gray mold), and the vines were trained using a Guyot system. The within- and between-row spacing were 1.0 and 2.3 m, respectively. Powdery and downy mildews were controlled according to an integrated pest management program, and the applied fungicides were not effective against *B. cinerea*. Ripe berries with no visible injuries or signs of *B. cinerea* infection were collected (with their pedicels) in 2018 and 2019 on September 3, 10, and 17 in 2018 and August 26 and September 2 and 9 in 2019. The ripening stages of the berries on the sampling dates were measured as degrees Babo (an indicator of sugar content) and were 19.3, 20.0, and 20.3 in 2018 and 16.4, 18.6, and 19.0 in 2019, respectively. Berries were transported to the laboratory in a cooler, after which they were rinsed under tap water for 3 min, disinfected with 2/3 of distilled water and 1/3 of 5% sodium hypochlorite to remove epiphytic microflora, and finally rinsed again with sterile water.

To study the effects of temperature on BCA efficacy (Exp 3), the berries were immersed for 30 s in beakers containing the BCA suspensions obtained as described before and continuously shaken. Berries immersed in double-distilled sterile water were used as non-treated (NT) controls. The berries were then placed in metal boxes (20 × 15 cm) over a metal grid net so that berries did not touch each other or the box bottom. Fifteen berries were placed into each box. After treatment, boxes were incubated in growth chambers with a 12 h photoperiod at 15, 20, 25, and 30 °C. After 1, 3, 6, 9, or 13 days, berries were inoculated with a conidial suspension of *B. cinerea*, isolate 213T, which belongs to the transposa sub-species and is highly aggressive [30]. The *B. cinerea* conidia were obtained from 10-day-old cultures grown on PDA at 20 °C and with a 12 h photoperiod using white and near-ultraviolet (UV) light (UV-A at 370 nm; Black Light UV-A, L18 w/73, OSRAM, Munich, Germany). The conidial suspensions were prepared by flooding the dishes with sterile-distilled water and gently scraping the agar surface with a sterile rod. The suspension was passed through two layers of sterilized gauze (autoclaved at 120 °C for 20 min.), and the conidia were enumerated with a hemocytometer. The concentration of conidia was adjusted to 10^5^/mL by adding double-distilled sterile water. The conidial suspension was uniformly distributed on the berries using a hand sprayer with 1 mL of suspension per box.

After inoculation with *B. cinerea*, boxes were placed in growth chambers at 25 °C, 100% relative humidity, and 12 h photoperiod for 7 days to promote the germination of *B. cinerea* conidia and the development of gray mold. Gray mold severity was then assessed as the percentage of the surface of each berry with gray mold symptoms, after which the severity of the entire replicate (consisting of the 15 berries in each box) was then determined by using the standard area diagram of Hill et al. [83].

Three boxes per temperature regime, incubation time, and berry growth stage were used. The experiment was performed three times per year for 2 years. Gray mold severity data were subjected to a factorial analysis of variance (ANOVA) in which the factors were BCA treatment (four BCAs plus the untreated control, NT), temperature (15, 20, 25, and 30 °C), and the number of days that berries were incubated before they were inoculated with *B. cinerea* (the colonization period of 1, 3, 6, 9, or 13 days). The experimental design was a split-split with BCAs as the main plots, temperature regimes as the split plots, and BCA colonization periods as the split-split plots. The 2 years and the different ripening stages during which berries were sampled were considered blocking factors. The gray mold severity data (in %) were subject to arcsin transformation before applying the ANOVA so that the variances were homogeneous.

To assess the variability in gray mold severity as affected by temperature and colonization period, CVs were calculated as described before. Gray mold severity data were then expressed as disease control relative to NT. The difference between gray mold severity on the untreated berries (the berries that were not treated with any BCA before they were inoculated with *B. cinerea*) and treated berries (the berries that were treated with a BCA before they were inoculated with *B. cinerea*) was calculated and divided by the maximal difference found in the experiment for the specific BCA [29].

## Figures and Tables

**Figure 1 plants-12-03430-f001:**
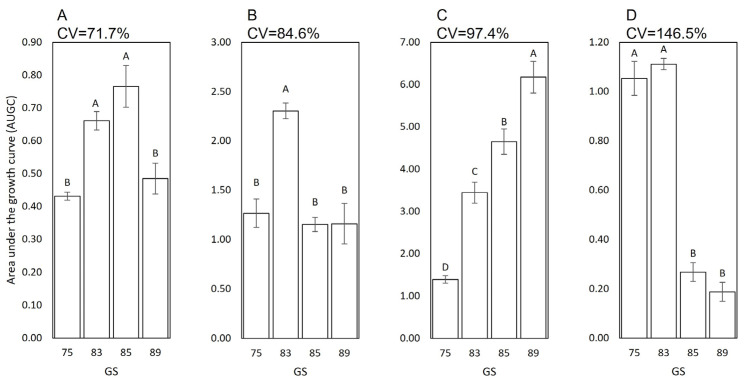
Area under the growth curve (AUGC) of four biocontrol agents (BCAs): (**A**): *Bacillus amyloliquefaciens* subsp. plantarum, *Aureobasidium pullulans*, *Metschnikowia fructicola*, and *Trichoderma atroviride* ((**A**): BAD, (**B**): APD, (**C**): MFN, and (**D**): TAS, respectively) on four artificial substrates that mimic the chemical composition of berries at different growth stages (GS75, 83, 85, and 89 corresponding to pea-sized, veraison, softening, and ripe berries, respectively). The BCA growth data were obtained after the inoculation and incubation of each substrate in the BioTek 800 TS absorbance reader at 20 °C for 48 h. CV represent the coefficient of variation of AUGC. Bars represent means, and whiskers indicate standard errors. Significantly different values within each BCA are indicated by different letters according to the Tukey’s test at α = 0.05.

**Figure 2 plants-12-03430-f002:**
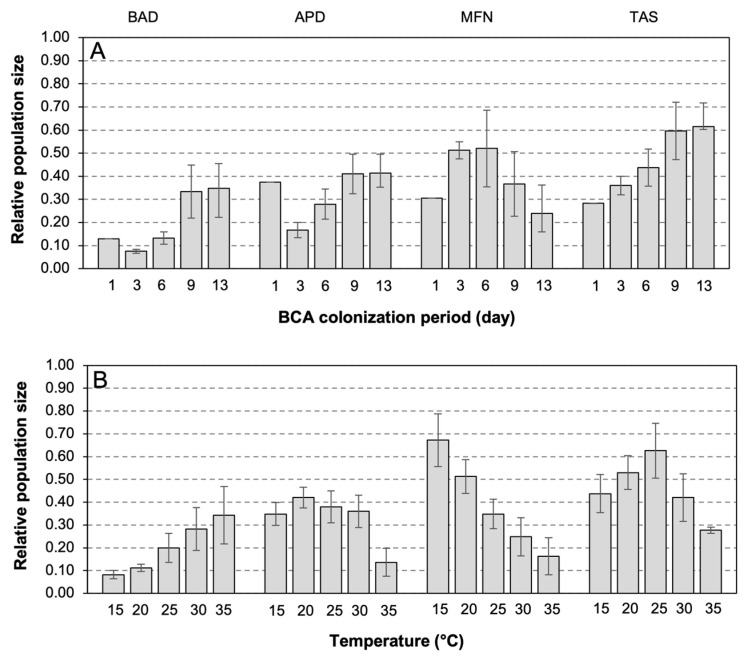
Relative population size of four BCAs (BAD, APD, MFN, TAS; see Table 1 for acronyms) on artificial substrates mimicking grape berries’ composition at ripening (GS89) as affected by (**A**) the length of the colonization period (in days) and (**B**) the temperature during the colonization. Bars are overall means of different temperature in (**A**) and of different numbers of days after the BCA inoculation in (**B**); whiskers indicate standard error. Population size was estimated as colony forming units (CFUs), and relative values were calculated by dividing each CFU value by the maximum of each BCA.

**Figure 3 plants-12-03430-f003:**
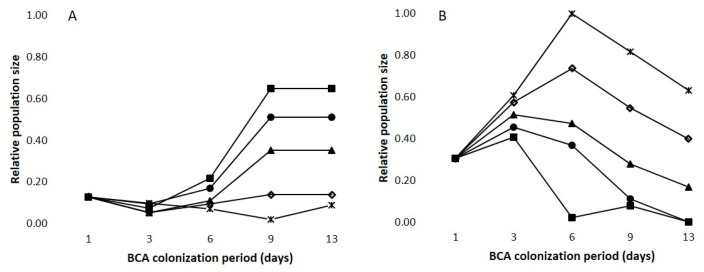
Relative population size on artificial substrates mimicking grape berries’ composition at ripening (GS89) of two BCAs, BAD (**A**) and MFN (**B**) as affected by the length of the colonization period (in days) at several temperature regimes: 15 °C (×); 20 °C (◊); 25 °C (▲); 30 °C (●), and 35 °C (■).

**Figure 4 plants-12-03430-f004:**
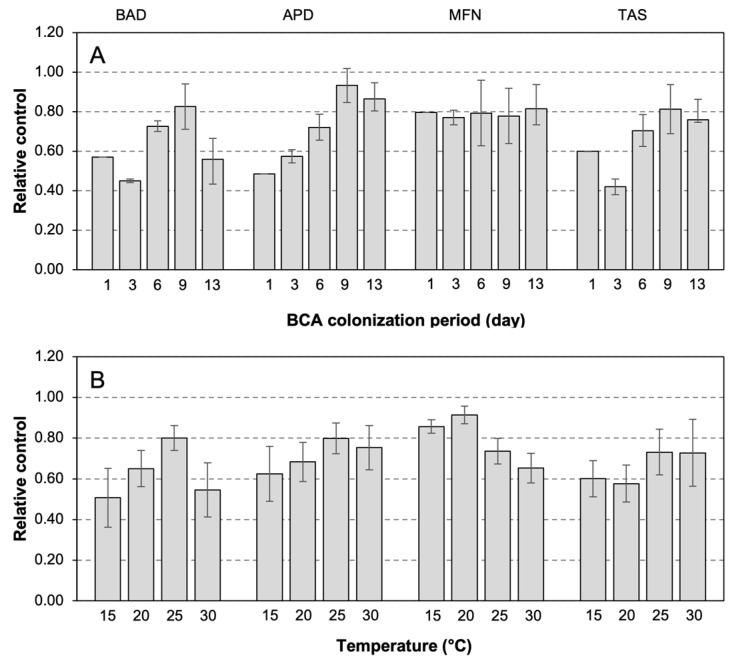
Relative control of gray in ripe grape berries by four BCAs (BAD, APD, MFN, TAS; see Table 1 for acronyms) as affected by (**A**) the length of the colonization period (in days) and (**B**) the temperature between BCA treatment and *B. cinerea* inoculation. Bars represent overall means of different temperatures in (**A**) and of different numbers of days after BCA inoculation in (**B**); whiskers indicate standard error. Relative control was calculated as the difference between gray mold severity on untreated and treated berries divided by the maximum difference found in the experiment for a specific BCA.

**Table 1 plants-12-03430-t001:** Biocontrol agents (BCAs) used in the experiment.

Active Ingredient	Commercial Product (Acronym)	Active Ingredient Concentration ^a^	Producer	Label Dose (g/ha)
*Bacillus amyloliquefaciens* D747	Amylo-X (BAD)	5 × 10^10^ CFU/g	CBC S.r.l., Bergamo (Italy)	2000
*Aureobasidium pullulans* DMS 14941-14940	Botector (APD)	2.5 × 10^9^ CFU/g (DMS 14941-14940)	Manica S.p.A., Trento (Italy)	400
*Metschnikowia fructicola* NRRL Y-27328	Noli (MFN)	1–3 × 10^10^ cells/g	Koppert Italia, Verona (Italy)	2000
*Trichoderma atroviride* SC1	Vintec (TAS)	1 × 10^13^ CFU/granule	Belchim S.p.A., Milan (Italy)	200

^a^ CFU/g = number of colony forming units per gram of product; cells/g = number of cells per gram of product; CFU/granule = number of colony forming units per granule of product.

## Data Availability

The raw data supporting the conclusions of this article will be made available by the authors without undue reservation.

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
