# Peer review of "Biocontrol of Botrytis cinerea as Influenced by Grapevine Growth Stages and Environmental Conditions"

_plants, 2023, doi:10.3390/plants12193430_

Round 1

Reviewer 1 Report

The manuscript "Biocontrol of Botrytis cinerea as influenced by grapevine growth stages and environmental conditions" presents interesting research results, but requires corrections before publication.

The manuscript presents the effects of commercial biocontrol agents on the growth inhibition of B. cinerea. This is an interesting topic due to the potential possibility of reducing the use of pesticides in plant production.

Detailed notes:

There should be more results in the abstract.

The figures lack statistical analysis.

What strain of B. cinerea was used? If the isolate, how was it identified?

line 428 - no superscript in spore concentration.

Author Response

Detailed notes:

There should be more results in the abstract.

A: modified

The figures lack statistical analysis.

A: In figure 2 and 4, we were interested in showing the response of BCAs to varying conditions of variables that are continuous by nature (i.e. time and temperature); therefore, the levels we used (1,3, .. days; 15, 20 .. °C, etc.) may be considered only samples of many possible values we could extract from the continuous variable. In such a situation, the post-hoc test is not usually used (and for some experts it is also incorrect). In the present study, the ANOVA was used only for understanding the significant factors in the experiment and their relative weight (as explained variance) and not for analyzing the differences between levels. It is not interesting to know if 15°C is different from 20°C, because then the difference between 16/17°C and 20°C should be also verified (and this create almost unlimited possibilities).

What strain of B. cinerea was used? If the isolate, how was it identified?

A: this information was reported in line 420: isolate 213T, which belongs to the transposa subspecies and is highly aggressive (Ciliberti et al., 2016).

line 428 - no superscript in spore concentration.

A: corrected

Reviewer 2 Report

This is an interesting and nicely written manuscript emphasizing the influence of the environment on the activity of BCA. Although it is widely accepted that the activity of biocontrol products based on live microorganisms, is highly dependable on the environment, experimental data on the subject are scarce. This article nicely fills the knowledge gap. The study is properly planned and well described. The results are analyzed properly and support the conclusions. My only concern is that the abstract does not fully describe the content of the manuscript. In my opinion, the abstract should contain more general information describing the significance of this research, pinpoint the most important results, and present the major conclusions. I would also appreciate if the authors could include the raw supporting data in the supplementary data.

Generally, I consider this manuscript suitable for publication in Plants after minor revisions.

In-text comments:

Please consider rewriting the abstract. Currently, the abstract describes the used methodology, please describe the idea behind the research and the meaning of the results.

Line 376: P should be exchanged for α

Please exchange Figure 3 for a higher-resolution graphic. 

Author Response

In-text comments:

Please consider rewriting the abstract. Currently, the abstract describes the used methodology, please describe the idea behind the research and the meaning of the results.

A: modified

Line 376: P should be exchanged for α

 A: modified

Please exchange Figure 3 for a higher-resolution graphic. 

A: replaced